# Re-assessing thermal response of schistosomiasis transmission risk: Evidence for a higher thermal optimum than previously predicted

**Ibrahim Halil Aslan**[1,2]*, **Julie D. Pourtois**[1,2], **Andrew J. Chamberlin**[2], **Kaitlyn R. Mitchell**[1,2], **Lorenzo Mari**[3], **Kamazima M. Lwiza**[4], **Chelsea L. Wood**[5], **Erin A. Mordecai**[1,6], **Ao Yu**[7], **Roseli Tuan**[8], **Raquel Gardini Sanches Palasio**[9], **Antônio M. V. Monteiro**[10], **Devin Kirk**[1], **Tejas S. Athni**[1,11], **Susanne H. Sokolow**[1,6], **Eliezer K. N'Goran**[12], **Nana R. Diakite**[12], **Mamadou Ouattara**[12], **Marino Gatto**[3], **Renato Casagrandi**[3], **David C. Little**[13], **Reed W. Ozretich**[13], **Rachel Norman**[14], **Fiona Allan**[15], **Andrew S. Brierley**[16†], **Ping Liu**[4], **Thiago A. Pereira**[17], **Giulio A. De Leo**[1,2]

**1** Department of Biology, Stanford University, Stanford, California, United States of America, **2** Hopkins Marine Station, Stanford University, Pacific Grove, California, United States of America, **3** Department of Electronics, Information and Bioengineering, Politecnico di Milano, Milano, Italy, **4** School of Marine and Atmospheric Sciences, Stony Brook University, New York, New York, United States of America, **5** School of Aquatic and Fishery Sciences, University of Washington, Seattle, Washington, United States of America, **6** Woods Institute for the Environment, Stanford University, Stanford, California, United States of America, **7** Department of Earth System Science, Stanford University, Stanford, California, United States of America, **8** Pasteur Institute, São Paulo Health Public Office, São Paulo, Brazil, **9** Department of Epidemiology, School of Public Health, University of São Paulo, São Paulo, Brazil, **10** National Institute for Space Research, São José dos Campos, São Paulo, Brazil, **11** Harvard Medical School, Boston, Massachusetts, United States of America, **12** Université Félix Houphouët-Boigny, Abidjan, Côte d'Ivoire, **13** Institute of Aquaculture, University of Stirling, Stirling, United Kingdom, **14** Computing Science and Mathematics, University of Stirling, Stirling, United Kingdom, **15** Department of Life Sciences, Natural History Museum, London, United Kingdom, **16** Scottish Oceans Institute, School of Biology, University of St. Andrews, St. Andrews, United Kingdom, **17** Institute for Stem Cell Biology and Regenerative Medicine, School of Medicine, Stanford University, Stanford, California, United States of America

† Deceased.
* iaslan@stanford.edu

**Data Availability Statement:** Readers can access the code and the literature review data at https://

## Abstract

The geographical range of schistosomiasis is affected by the ecology of schistosome parasites and their obligate host snails, including their response to temperature. Previous models predicted schistosomiasis' thermal optimum at 21.7˚C, which is not compatible with the temperature in sub-Saharan Africa (SSA) regions where schistosomiasis is hyperendemic. We performed an extensive literature search for empirical data on the effect of temperature on physiological and epidemiological parameters regulating the free-living stages of *S. mansoni* and *S. haematobium* and their obligate host snails, i.e., *Biomphalaria* spp. and *Bulinus* spp., respectively. We derived nonlinear thermal responses fitted on these data to parameterize a mechanistic, process-based model of schistosomiasis. We then re-cast the basic reproduction number and the prevalence of schistosome infection as functions of temperature. We found that the thermal optima for transmission of *S. mansoni* and *S. haematobium* range between 23.1–27.3˚C and 23.6–27.9˚C (95% CI) respectively. We also found that the

github.com/ibrahimhalilaslan/Thermal_sensitive_schistosomiasis_model.

**Funding:** GADL, IHA, AJC, KRM, JDP have been partially supported by NSF (https://www.nsf.gov/) (CEH/NSF ICER-202483 and NSF-EEID DEB 2011179). GADL, IHA, AJC, KRM, JDP have been partially supported and EAM was funded by Stanford University Center for Innovation in Global Health (https://globalhealth.stanford.edu/, USA). GADL, IHA, AJC, KRM, JDP have been partially supported and EAM was funded by Stanford Woods Institute for the Environment (https://woods.stanford.edu/). EAM was funded by NSF (https://www.nsf.gov/) (DEB-2011147 with Fogarty International Center) and the National Institutes of Health (R5GM133439, R01AI168097, R01AI102918). TAP was supported by the National Institute of General Medical Sciences (T32GM144273). DCL, RWO, RN, FA, ASB were supported by NERC (https://www.ukri.org/councils/nerc/, UK) (NE/T01710/1). EKN, NRD were funded by PASRES (https://www.csrs.ch/pasres/, Côte d'Ivoire) (19/2593). RT, RGSP and AMVM was funded by FAPESP (https://fapesp.br/en, Brazil) (2019/2593-3). The funders had no role in study design, data collection and analysis, decision to publish, or preparation of the manuscript.

**Competing interests:** The authors have declared that no competing interests exist.

thermal optimum shifts toward higher temperatures as the human water contact rate increases with temperature. Our findings align with an extensive dataset of schistosomiasis prevalence in SSA. The refined nonlinear thermal-response model developed here suggests a more suitable current climate and a greater risk of increased transmission with future warming for more than half of the schistosomiasis suitable regions with mean annual temperature below the thermal optimum.

## Author summary

In this research, we explored the complex interplay between temperature and the transmission risk of schistosomiasis, a parasitic disease currently affecting over two hundred million people, predominantly in SSA. We developed a novel mathematical model accounting for the multiple positive and negative ways temperature affects the free-living stages of the parasite and its obligate, non-human host, i.e., specific species of freshwater snails. Our models show that schistosomiasis transmission risk peaks at temperatures 1–6°C higher than previously estimated. This indicates that the impact of climate change on schistosomiasis transmission might be more extensive than previously thought, affecting a wide geographic range where mean annual temperatures are currently below the optimal temperature. Our model projections are consistent with the observed temperatures in locations of SSA where schistosomiasis is endemic and data on infection prevalence in the human population are available. These findings suggest that the current climate is conducive to schistosomiasis transmission, and future warming could escalate the risk further, emphasizing the need for targeted interventions in these regions.

## Introduction

The Earth's temperature has increased by approximately 1.5 degrees Celsius over the past 100 years and some climate change scenarios project even faster increases in the future [1–4], with pervasive impacts on ecosystems and human health. Temperature is one of the most important environmental determinants of the life history traits (LHT) of living organisms [2,5–9], especially ectotherms. Many of these ectotherms are involved in the transmission of environmentally mediated diseases in humans, making transmission temperature-dependent [7]. Schistosomiasis is one of these environmentally mediated diseases. It is a water-associated, acute and chronic parasitic disease of poverty, infecting more than two hundred million people worldwide, the vast majority of whom are in SSA [10,11].

 Schistosomiasis (also known as bilharzia or snail fever) is caused by parasitic flatworms of the genus *Schistosoma*. Of the five *Schistosoma* species of public health importance—*S. mansoni*, *S. haematobium*, *S. japonicum*, *S. intercalatum* and *S. mekongi*—most of the burden of schistosomiasis is caused by *S. mansoni*, endemic in both Africa and South America, and *S. haematobium*, endemic in Africa. These parasites have a complex life cycle that includes intermediate host snails and free-living stages, miracidia and cercariae, which are ectotherms (unbale to regulate their temperature) and sensitive to environmental conditions. In brief, humans with an active infection pass *Schistosoma* spp. eggs through their urine (*S. haematobium*) or stools (*S. mansoni*), and the eggs hatch upon water contact. The larvae, called miracidia, seek and infect freshwater host snails (of the genera *Bulinus* and *Biomphalaria* for *S. haematobium* and *S. mansoni*, respectively) and asexually reproduce in snail tissues. A few

weeks after snail infection, depending upon temperature, snails start shedding a second free-living stage of the *Schistosoma* parasite, called cercariae, which infect humans by penetrating through the skin. After a month, the parasites move in the blood plexus of the human bladder (*S. haematobium*) or the small intestine (*S. mansoni*) where they reproduce sexually [12–15].

Temperature is a crucial driver at every stage of the schistosomiasis life cycle outside the human body and affects different species of *Schistosoma* and their host snails differently. The free-living stages (cercariae and miracidia) are directly impacted by water temperature. Effects of temperature have been reported for the survival and infectivity of cercariae and miracidia [16–25], mortality rate of cercariae [18,26], miracidia hatching, and cercariae emergence [27], snail fecundity, growth, mortality [10,28–30,30–34], and parasite incubation period within the snail [15,21,35,36], demonstrating pervasive effects of temperature across the schistosome life cycle. While these experiments provide insight on the response of individual LHTs, understanding the impact of temperature on the entire *Schistosoma* life cycle requires the integration of these LHTs across all snail and parasite life stages. Mathematical models [6,37,38] provide us with the flexibility and power to combine these LHTs and thereby predict how transmission and infection prevalence will be affected by temperature change.

Several mathematical models have been developed to examine the thermal response of schistosomiasis transmission [27,39–41]. However, these models are typically limited to a particular life stage of the parasites [27] or omit the temperature dependence of important LHTs [39,41]. For example, the model presented in [42] and later used by [27] does not include the prepatent period in snails, which is strongly correlated with temperature; furthermore, the temperature dependence of snail egg production, hatching, and survival rate have not previously been incorporated. In addition, the differences in the temperature response between *S. mansoni* and *S. haematobium* [15,21,35] have not always been considered in previous studies [27,39,41,43,44], nor how temperature might affect frequency or intensity of human contact rate with water [45]. Thus, we still lack a comprehensive understanding of the impact of temperature on schistosomiasis transmission that fully accounts for its complex life cycle and differences between *Schistosoma* species. We developed a comprehensive thermal-sensitive mechanistic model of schistosomiasis and tested its performance across SSA using data from the Global Neglected Tropical Disease (GNTD) database [46,47]. The temperature-dependent parameters used in the model are based on thermal performance curves (TPCs) that were fitted separately to empirical data for *S. mansoni* and *S. haematobium* to account for their unique LHTs. The species-specific thermal responses of the basic reproduction number for schistosomiasis ($R_0$) and the prevalence of infection in humans were calculated for *S. mansoni* and *S. haematobium* to measure the intensity of transmission with respect to temperature. We then identified the most critical temperatures, the confidence intervals around the $R_0$ and the prevalence curves for both parasites. We also analyzed the impact of human water contact behavior on the thermal response of schistosomiasis transmission. We compared our estimates of the optimal temperature for schistosomiasis transmission with previous published estimates by using data from the GNTD database. We showed that the thermal response of $R_0$ and the prevalence in humans are unimodal curves, and that their thermal optimum is higher than previously determined [27,40]. We also found that an increase in the human water contact rate in response to increasing temperature, as expected under climate change scenarios, can also shift the thermal optimum towards higher values. Finally, we mapped temperature-dependent $R_0$ across Africa for each parasite and identified regions of the continent where temperature is either below or above the thermal optimum, i.e., regions where a temperature increase due to climate change might either increase or decrease transmission risk, respectively. We found that a large fraction of the continent might experience an increase in schistosomiasis transmission risk as a consequence of climate change.

## Materials and methods

### Model description

We built a system of ordinary differential equations (ODEs) with temperature-dependent parameters to formulate a mechanistic model of schistosomiasis transmission. The model consists of three compartments for snails and two compartments for worms inside the human body (Fig 1): susceptible snails ($S$); prepatent snails, infected by miracidia but not shedding cercariae yet, which is also the stage of developing sporocysts [48] ($P$); infectious snails, shedding cercariae ($I$); mean number of immature worms that do not produce eggs ($W$); and mean number of mature worms that produce eggs ($W_m$).

The dynamics of free living cercariae ($C$) and miracidia ($M$) are considered implicitly: the lifespan of miracidia and cercariae is short [16–18,26] (i.e., less than 1 day) with respect to the lifespan of snails and adult parasites; we therefore embedded their faster dynamics into the slower dynamics of the rest of the model by assuming that the abundances of the larval stages are at quasi-equilibrium with the other state variables for a given temperature, that is

$$M^* = \frac{h v_e \delta_e(T)}{\mu_m(T)} W_m \tag{1}$$

$$C^* = \frac{v_c(T)}{\mu_c(T)} I \tag{2}$$

where $T$ is water temperature (referred just as temperature in the rest of this study), $h$ is the

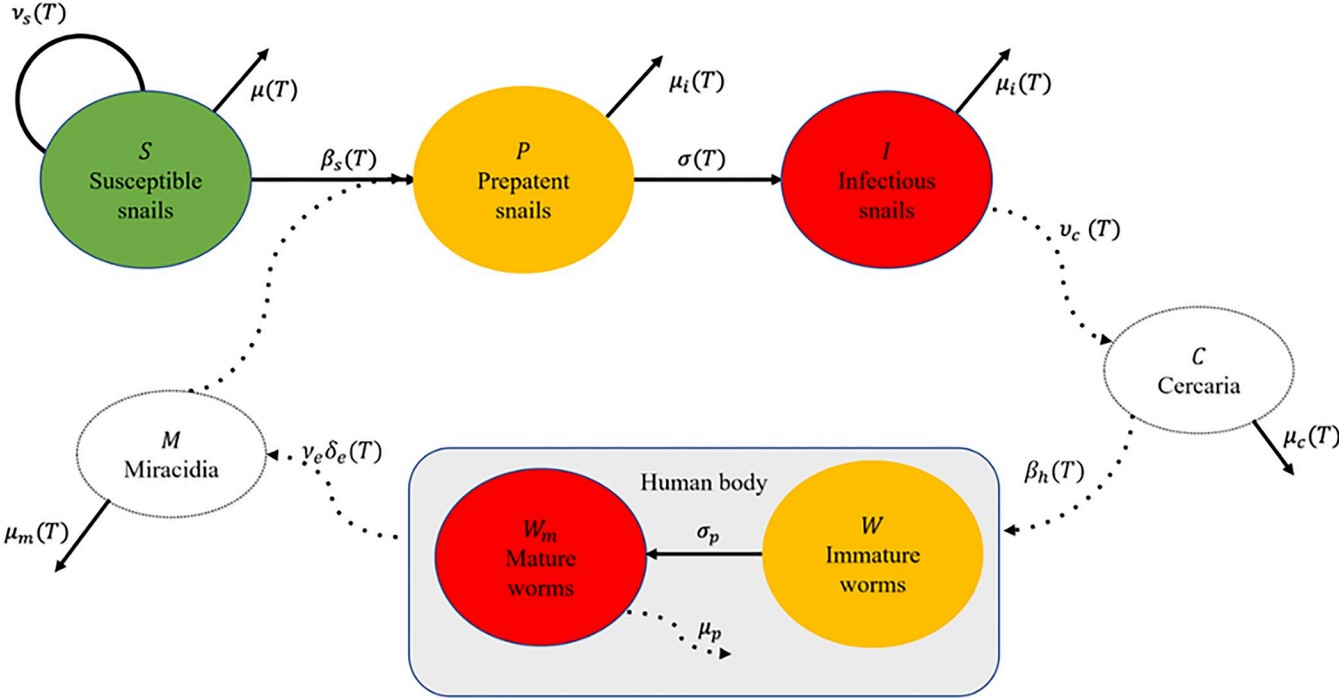

**Fig 1.** The transmission of schistosomiasis requires infection of a susceptible snail by free-living miracidia and the subsequent release of cercariae in the water after a prepatent period. Cercariae can infect humans, where they become mature worms and produce eggs. Eggs are then released from the human body through urine or stool and later hatch and become miracidia, thus completing the parasite's life cycle. Processes whose dynamics are considered explicitly are shown with a solid arrow, while processes for which quasi-equilibrium is assumed are depicted with a dashed arrow. We highlight uninfected compartments with green, infected compartments with red, and incubation periods with yellow.

human population size (assumed constant), $v_e$ is the number of eggs produced daily by a mature worm, $\delta_e(T)$ is the probability that eggs successfully hatch into miracidia, $\mu_m(T)$ is the mortality rate of miracidia, $v_c(T)$ is the per-capita daily *cercaria* shedding rate, and $\mu_c(T)$ is the mortality rate of cercariae (see Figs A-I in S1 Supplementary Materials). The model describing schistosomiasis dynamics is thus represented by the following set of ODEs:

$$\frac{dS}{dt} = (v_s(T) - v(S + P + I))(S + rP) - \lambda(T)S - \mu(T)S \tag{3}$$

$$\frac{dP}{dt} = \lambda(T)S - (\sigma_S(T) + \mu_i(T))P \tag{4}$$

$$\frac{dI}{dt} = \sigma_S(T)P - \mu_i(T)I \tag{5}$$

$$\frac{dW}{dt} = C_w(T)\beta_h(T)C^* - \sigma_p W \tag{6}$$

$$\frac{dW_m}{dt} = \sigma_p W - \left(\mu_h + \mu_p\right)W_m \tag{7}$$

Snails reproduce at the per-capita rate $v_s(T)$, reduced by $v$ ($S+P+I$) to account for (logistic-like) density-dependent competition among snails (Eq 3). The fecundity of infected but not infectious (i.e., prepatent) snails is reduced by a factor $r$ due to the infection, whereas snails in the *I* compartment do not reproduce [49]. Following [11], the per-capita force of infection $\lambda(T)$, i.e., the rate at which susceptible snails become infected, is assumed to be an increasing and saturating function of the number of miracidia $M^*$ relative to that of snails $N = S+P+I$, namely:

$$\lambda(T, t) = \Lambda\left(1 - e^{-\beta_s(T)\frac{M^*}{N}}\right) \tag{8}$$

where $\Lambda$ is the maximum force of infection and $\beta_S(T)$ is the transmission rate from miracidia to snails. Prepatent snails become infectious at a rate $\sigma_S(T)$, and prepatent and infectious snails both die at a rate $\mu_i(T)$ (Eqs 4 and 5). Uninfected snails die at a per-capita rate $\mu(T)$. $\beta_h(T)$ is the transmission rate of schistosomiasis in human population (Eq 6). $C_w(T)$ modulates the human contact rate with water in response to temperature, following observations that the frequency and duration of contact with waters are higher at higher temperatures [45]. Finally, worms mature at a rate $\sigma_p$ and die at a rate $\mu_p+\mu_h$, where $\mu_h$ represents the death rate of the human host and $\mu_p$ is the mortality rate of the worm itself (Eq 7). The maturation rate of the parasite, $\sigma_p$, the egg production rate, $v_e$, and the death rates of worms, $\mu_p$, do not depend on external temperatures because these life stages occur in the human body at a fixed temperature. The human mortality rate, $\mu_h$, is also assumed to be independent of temperature.

## Model parameterization

We conducted a literature review of peer-reviewed papers reporting empirical data on thermally sensitive model parameters derived from laboratory experiments. We restricted our search to experiments and data for *S. mansoni* and *S. haematobium* parasites, and on snails of the genus *Biomphalaria* and *Bulinus* spp., respectively, without any time or geographical constraint. We used the search engine of ISI Web of Science between Nov 2022 and April 2023 and found 138 papers matching the keywords: ("*temperature*" OR "*thermal response*" OR

"thermal sensitivity") AND schistosomiasis AND snail. We identified about 50 additional papers reporting experimental results and used their bibliography and citations to identify further papers that might have not been included in the original search. When necessary, demographic rates were converted to the same unit of our model (see S1 Supplementary Materials); the data regarding the transmission rates $\beta_s(T)$ and $\beta_h(T)$ were also converted to the units of our model by using an SIR type model (S1 Supplementary Materials). For each demographic and epidemiological thermally sensitive parameter, we compiled the data collected from multiple empirical studies (when available) and estimated the best-fitting curve with the R package rTPC (Padfield and O'Sullivan) [50]. rTPC was specifically developed to fit a family of 24 TPCs previously described in the literature (e.g., quadratic, Brière, Gaussian) using non-linear least squares regression. This R package requires at least 7 data points to fit the family of 24 curves. We determined the best-fitting curve for each temperature-dependent parameter out of the family of 24 curves based on the Akaike Information Criterion (AIC), number of parameters, and biological applicability. We disregarded AIC criteria when the curve with lowest AIC exhibited unrealistic trends, such as negative values in the range of temperatures or a multimodal behavior near extreme temperatures in schistosomiasis transmission areas. TPCs were estimated separately for *S. mansoni* and *S. haematobium*, as well as for their corresponding host snails at the genus level, i.e., *Biomphalaria* and *Bulinus* spp., respectively. To determine the uncertainty around each TPC, we used a bootstrap resampling method to construct the 95% confidence intervals by using the rTPC. Results of the bootstrap resampling are shown in the S1 Supplementary Materials. Temperature-invariant parameters were estimated from the literature [13,39,51]. The human population $h$ was set to 1,000 people, and the parameters $\nu$ and $\Lambda$ [13,51–53] were fit so that prevalence of infection in snails is around 5%, as documented by [13]. In addition, we adjusted these parameters to set the Mean Parasite Burden (MPB) in the human population approximately to be 80 [44,54].

While it is expected that frequency and duration of human contact with water may increase with temperature, there is a paucity of rigorous empirical studies systematically reporting field observations that can be useful to estimate the water contact rate as a function of temperature, as documented for instance in [45]. For these reasons, we first ran simulations assuming that the relative contact rate with water $C_w(T)$ is constant and equal to 1, and derived the corresponding thermal envelope and the thermal optima. Then, we ran a further set of simulations assuming that the contact rate is an increasing and saturating function of temperature that levels off when temperature is low, namely:

$$C_w(T) = \frac{1}{1 + e^{-a(T - T_{med})}} + b \tag{9}$$

where:

$b \geq 0$ is the minimum contact rate at low temperature

$T_{med}$ is the temperature at which the slope of mid-point occurs

$a \geq 0$ is the parameter that governs the steepness of the function.

This curve was qualitatively parameterized by using field observations reported by Sow et al. (2011) [45].

## Basic reproduction number $R_0$ and prevalence in human

The basic reproduction number $R_0$ is the number of new infections caused by one infectious case in a fully susceptible population [55] and is generally used to determine the stability of the disease-free equilibrium (DFE) [37]. If $R_0 < 1$, the DFE is stable and the disease is not able to invade and establish in the population; otherwise, the DFE is unstable, the disease is able to

establish in the population, and the number of infected snails and the MPB in the human population may converge toward a non-trivial (i.e., strictly positive), long-term equilibrium. Note that we are only considering parameter sets in here for which the global extinction equilibrium is asymptotically unstable. We derived the basic reproduction number $R_0$ by using the next generation matrix method [55,56] (see S1 Supplementary Materials for detail).

$$R_0(T) = \left( \frac{\Lambda \beta_s(T) h \delta_e(T) \nu_e \beta_h(T) \nu_c(T) \sigma_s(T)}{\mu_m(T)(\mu_h + \mu_p)\mu_c(T)\mu_i(T)(\sigma_s(T) + \mu_i(T))} \right)^{1/2} \tag{10}$$

where the parameters are either constant or a function of temperature $T$.

The basic reproduction number $R_0$ is particularly useful to assess how likely a disease can invade a fully susceptible population. In addition, a temperature-dependent $R_0$ can inform about both thermal optimum and thermal breadth. On the other hand, schistosomiasis is also a chronic disease, with the adult parasite able to live for years in the human body. Therefore, it is also interesting to assess the relationship between temperature and the MPB at the endemic equilibrium. [46] Accordingly, we simulated disease dynamics until the model reached the endemic equilibrium, and, under the common assumption of a negative binomial distribution of the adult parasites in the human population [57], we used the mean parasite burden at equilibrium to derive the prevalence of infection in the human population (specifically, the fraction of individuals hosting one or more pairs of worms) as a function of temperature.

## Model comparison with prevalence data

We used the GNTD database to evaluate whether our model results are consistent with the prevalence data observed in the field. The 2015 version of the GNTD database included 11,333 *S. mansoni* data points, of which 6,250 have non-zero schistosomiasis prevalence in the human population, and 14,313 *S. haematobium* data points, of which 4,873 have non-zero prevalence, out of nearly 8,000 unique locations. We limited our analysis to locations with non-zero prevalence, as our model parameterization assumes that schistosomiasis is present in the population. We then used quantile regressions to investigate whether the $R_0$ estimates obtained from our model performed better or worse as a predictor of reported prevalence than the estimates from the thermally sensitive model presented in [27]. Models were compared using AIC. For ease of visualization, we created bins made of 400 data points and showed the 98th percentile of the bins. Note that GNTD data are divided by their maximum value and multiplied by the maximum value of the corresponding curve to make visually comparable. Details about the models and visualizations of the raw dataset are available in the S1 Supplementary Materials.

## Projection of $R_0$ across Africa

We use the thermal sensitive curve $R_0$ to map the relative value of $R_0$ as a function of temperature across Africa. Location specific mean annual temperatures were derived from [58–60] over a grid with a 0.5-degree (~50km) spatial resolution across the continent. Note that the temperature is a shortcut for water temperature in this study and assume that the waterbody temperature is equal to the mean annual temperature. As [61] found that large water bodies may affect schistosomiasis transmission up to a 20–30 km distance, we used a mask to remove areas that are more than 25km aways from permanent or temporary water bodies and with extremely low population densities, such as in desertic areas (see S1 Supplementary Materials). In addition to $R_0$, we also mapped $dR_0/dT$ across Africa i.e., the first derivate of $R_0$ as a function of temperature to identify the areas that are currently below the thermal optimum (and whose derivate is thus positive) and apart them from the areas that are already above the thermal

optimum (and whose derivate is thus negative), as in these areas we expect that an increase in temperature caused by climate change might increase or decrease schistosomiasis transmission risk. We used the R package "map" version 3.4.2 to obtain the outlines of countries in Africa and then plot by using the R package "ggplot2".

### Parameter sensitivity analysis

We conducted a sensitivity analysis to discern the impact of each parameter on the disease transmission dynamics. Briefly, we derived the partial derivative of $R_0$ with respect to temperature, while holding all other parameters constant except for the parameter of interest (details provided in S1 Supplementary Materials). In addition, we analyzed how much the thermal response of each demographic and epidemiological parameter influences the model outcomes by deriving the thermal response of $R_0$ while keeping the parameter of interest constant.

## Results

### Thermal performance of model parameters

TPCs were fitted for each temperature-dependent parameter in equations (Eqs 1–9). Data and best-fit curves are shown in **Fig 2** (see S1 Supplementary Materials for confidence intervals) and summarized in Table 1. Specifically, TPCs were estimated for snail fecundity rate, natural mortality, additional mortality caused by infection, and prepatent period for *Biomphalaria* and *Bulinus* spp. separately. TPCs for the cercaria shedding rate in snails were only fitted to data for *Biomphalaria* spp., as we could not find similar studies for *Bulinus* spp. TPCs for the mortality rate of miracidia and the duration of the prepatent phase were fitted separately for *S. mansoni* and *S. haematobium*, whereas TPCs for the probability of cercariae successfully infecting the human host, probability of miracidia hatching success, and mortality rate of cercariae were only fitted to data for *S. mansoni*.

We used the Euler-Lotka equation to derive the fecundity rate of *Biomphalaria* and *Bulinus* snails (**Fig 2**A and 2B), respectively, from snails' egg laying, hatching, survival, and maturation rates (S1 Supplementary Materials). Separate thermal curves for each of these processes are available in the S1 Supplementary Materials. We found that the best-fit function for the fecundity rate of *Biomphalaria* spp. (*B. pfeifferi*, *B. alexandrina*, *B. glabrata*, *B. sudanica*) is a concave unimodal shape that peaks at 24.0˚C, with a rough approximation of thermal breadth (14, 33˚C). On the other hand, the best fit for *Bulinus* spp. (*B. globosus*, *B. truncatus*, *B. nyassanus*) fecundity data is a Johnson Lewin curve with a peak close to 28˚C and a rough approximation of thermal breadth (10, 32˚C). These peak temperatures are higher than the previous estimate (22.1˚C) for all *Biomphalaria* and *Bulinus* spp. combined. Overall, we found *Bulinus* spp. to have a lower fecundity rate than *Biomphalaria* spp., with *Biomphalaria alexandrina* having the lowest fecundity rate among *Biomphalaria* spp. (see S1 Supplementary Materials). We also observed that the confidence interval around the fecundity rate TPC for *Biomphalaria* spp. is narrower than for *Bulinus* spp. (Panel B in Fis A-B in S1 Supplementary Materials).

The mortality rates of snails (**Fig 2**C and 2D) are calculated from the fraction of snails surviving during each laboratory experiment at a given temperature. We log-transformed the survival curve for the sake of visualization. The best fit for *Biomphalaria* spp. (*B. pfeifferi*, *B. alexandrina*, *B. sudanica*) is a unimodal convex curve (Spain) [62] with a minimum at 21˚C. The best fit for *Bulinus* spp. (*B. globosus*, *B. truncatus*, *B. africanus*, *B. nyassanus*) is a quadratic function [63] with a minimum close to 15˚C. The empirical data show that mortality increases rapidly at high temperatures. The best fit across the entire temperature range accounted for in this study was obtained by using two curves, one for temperatures below 33˚C for *Biomphalaria* or 36˚C for *Bulinus* (**Fig 2**C and 2D) and the other for temperatures above 33 or 36˚C

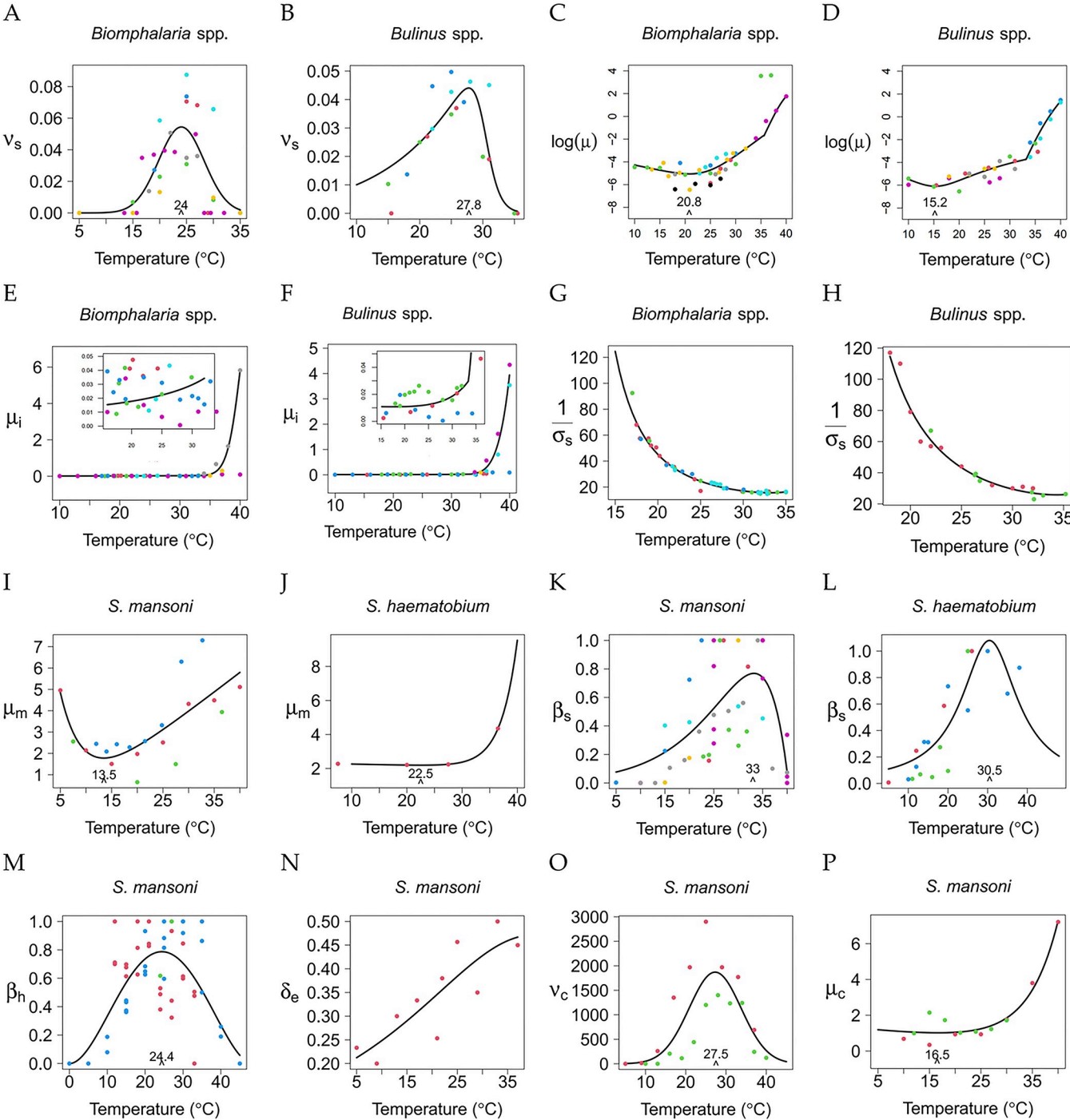

**Fig 2.** Thermal performance curves for temperature-dependent parameters. Different colors are used for different data sources. Parameters on the y-axes: $\nu_s$ snail fecundity rate; $\log(\mu)$ log transformation of the mortality rate of uninfected snails; $\mu_i$ mortality rate of infected snails, inset plot are zooming y-axis for the temperature range (15, 30°C); $\sigma_S(T)$ prepatent period in snails; $\mu_m(T)$ mortality rate of miracidia; $\beta_s(T)$, transmission rate from miracidia to snails; $\beta_h(T)$ transmission rate from cercaria to humans; $\delta_e(T)$ probability that eggs successfully hatch into miracidia; $\nu_c(T)$ per-capita daily cercarial shedding rate; $\mu_c(T)$ mortality rate of cercariae.

**Table 1. Summary of temperature-dependent parameters.**

| Parameter | Symbol in the model | Species | Shape | References | Fig 2 Panel | Optimal Temp. (95% CI) |
|---|---|---|---|---|---|---|
| Snail fecundity rate | $v_s(T)$ | *Biomphalaria* spp. | Unimodal Concave | [28–30,33,69,70] | A | 24.0°C (22.87, 25.60)°C |
| | | *Bulinus* spp. | Unimodal Concave | [10,29,34,70] | B | 27.8°C (25.56, 30.32)°C |
| Snail mortality rate | $\mu(T)$ | *Biomphalaria* spp. | Unimodal Convex | [21,28–30,32,69,71] | C | 21°C (17.8, 23.5)°C |
| | | *Bulinus* spp. | Unimodal Convex | [10,29,32,34,71,72] | D | 15°C (10, 19)°C |
| Infected snail mortality rate | $\mu_i(T)$ | *Biomphalaria* spp. | Increasing | [21,25,35,36] | E | NA |
| | | *Bulinus* spp. | Increasing | [10,20,73] | F | NA |
| Transition rate of snail from prepatency to infection | $\sigma_S(T)$ | *Biomphalaria* spp. | Decreasing | [15,21,35,36] | G | NA |
| | | *Bulinus* spp. | Decreasing | [15,73] | H | NA |
| Miracidia mortality rate | $\mu_m(T)$ | *S. mansoni* | Unimodal Convex | [16–18] | I | 13.5°C (11.36, 18.08)°C |
| | | *S. haematobium* | Unimodal Convex | [17] | J | 22.5°C Insufficient data |
| Transmission rate to snails | $\beta_s(T)$ | *S. mansoni* | Unimodal Concave | [16,17,21–25] | K | 33°C (31.51, 34.34)°C |
| | | *S. haematobium* | Unimodal Concave | [17,19,20] | L | 30°C (28.33, 33)°C |
| Transmission rate to humans | $\beta_h(T)$ | *S. mansoni* | Unimodal Concave | [18,22,23] | M | 24.4°C (22.27, 26.81)°C |
| Miracidia hatching success | $\delta_e(T)$ | *S. mansoni* | Increasing | [27] | N | NA |
| Number of cercariae released | $v_c(T)$ | *Biomphalaria* spp. | Unimodal Concave | [25,27] | O | 27°C (25.15, 31.10)°C |
| Mortality rate of cercariae | $\mu_c(T)$ | *S. mansoni* | Unimodal Conve | [18,26] | P | 16.5°C (10, 21.81)°C |

(for *Biomphalaria* and *Bulinus*, respectively). We observed lower variability for *Biomphalaria* spp. than *Bulinus* spp. in the confidence intervals around the TPC for the snail mortality rate (Panel B in Fis C-D in S1 Supplementary Materials).

The mortality rates of infected snails were calculated from the snail survival time at different temperatures after exposure to miracidia (**Fig 2**E and 2F). The best-fit curves for *Biomphalaria* spp. (*B. glabrata*, *B. pfeifferi*) and *Bulinus* spp. (*B. globosus*, *B. truncatus*) are Flinn [64] and Spain [62], respectively. The mortality rates increase as temperature increases for both *Biomphalaria* spp. infected with *S. mansoni* and *Bulinus* spp. infected with *S. haematobium*. As for the natural mortality rate of infected snails, we fitted the relevant TPCs to data for higher and lower temperatures separately (**Fig 2**E and 2F inset panels). Note that it was not possible to determine the temperature minimizing the mortality rate of infected snails due to significant variability in the data, particularly for *Biomphalaria* spp. (Table 1). This also leads to a wider confidence interval at higher temperatures (Panel B in Fis L-M in S1 Supplementary Materials).

The prepatent period, the average time between snail infection with miracidia and the onset of cercaria shedding (**Fig 2**G and 2H), is best fit by the right, positive branch of the Irf function [65] of temperature for both *Biomphalaria* spp. (*B. glabrata*, *B. pfeifferi*) and *Bulinus* spp. (*B. truncatus*, *B. globosus*) (Panel B in Fis E-F in S1 Supplementary Materials). Data show little variability in prepatent time among snails of the same genus, and that *Bulinus* snails have a longer prepatent time than *Biomphalaria* snails.

The mortality rate of miracidia is calculated from their survival time at different temperatures (**Fig 2**I and 2J). The best-fit curve is Thomas [66] for *S. mansoni*, with the lowest mortality rate at 13.5˚C, and Spain [62] for *S.haematobium*, with the lowest mortality rate occurring at 22.5˚C. Variability in the data led to a wide confidence interval around the TPC at high and low temperatures for *S. mansoni* (Table 1). Also, we could not calculate the confidence intervals around the TPC of *S. haematobium* through bootstrapping due to the small size of the dataset. In this case, we generated a confidence interval by resampling from the confidence interval of the estimated TPC parameters. The mortality rate of cercariae (**Fig 2**P) is also calculated from the proportion of surviving larvae at different temperatures. The best-fit curve for the data is Spain [62], with the lowest mortality of cercariae at 16.5˚C. We observe a rapid increase at high temperatures, with narrow confidence intervals around the TPC (Panel B in Fis I-K in S1 Supplementary Materials).

The transmission rate from miracidia to snails (**Fig 2**K and 2L) was estimated by utilizing a mechanistic model from data on the percentage of infected *Biomphalaria* spp. (*B. glabrata*, *B. pfeifferi*) and *Bulinus* spp. (*B. globosus*, *B. trancatus*) following exposure to *S. mansoni* and *S. haematobium*, respectively (see S1 Supplementary Materials). The best-fit curves are Spain [62] for *Biomphalaria* spp. exposed to *S. mansoni* and Flinn [64] for *Bulinus* spp. exposed to *S. haematobium*. The transmission rate peaked at 33˚C for *Biomphalaria* spp. and 30.5˚C for *Bulinus* spp. The transmission rate from cercariae to humans (**Fig 2**M) is estimated from a separate SIR model of mice and cercariae with the data on infection rate in the mice Experimental studies are only available for *S. mansoni*. The best-fit function for this data is a Brière curve [67], with an optimal temperature for transmission around 24.5˚C. We observe transmission happening at as low as 10˚C and as high as 40˚C. Due to different laboratory conditions, both the miracidia-to-snail and the cercaria-to-human transmission rates have strong noise in the data and wide confidence intervals (Panel B in Fis N-P in S1 Supplementary Materials).

The hatching probability of miracidia (**Fig 2**N) is evaluated as the proportion of miracidia successfully hatching. Only one study was found measuring this parameter and it focused on *S. mansoni*. The best-fit function for these data is Flinn [64], showing a monotonic increase with temperature. Uncertainty increases with temperature (Panel B in Fig G in S1 Supplementary Materials). The number of cercariae released per day by one snail (**Fig 2**O) was obtained from two empirical studies of *Biomphalaria* spp. infected by *S. mansoni*. Based on these two studies, the best-fit function is a Gaussian curve [68], with its peak at 27.5˚C, matching the finding in [27], and a rough approximation of thermal breadth (15, 40˚C). The confidence interval around the TPC becomes wider at higher temperatures (Panel B in Fig H in S1 Supplementary Materials).

### Thermal response of $R_0$ and prevalence in humans

To investigate how transmission intensity varies with temperature, we calculated the basic reproduction number $R_0$ and the prevalence of schistosomiasis in humans as discussed in the method section. We found that the thermal response curve of $R_0$ has a unimodal concave shape for both *S. mansoni* and *S. haematobium*, peaking at 25.5˚C (95% CI: 23.1–27.3˚C) and 26.2˚C (CI: 23.6–27.9˚C) with 3.68 and 3.44 peak value (Fig Q in S1 Supplementary Materials), respectively (**Fig 3**A and 3B). The thermal response of the prevalence in humans also has a concave shape for both *S. mansoni* and *S. haematobium*, with peaks at 24.0˚C (CI: 23.1–27.3˚C) for *S. mansoni* and 26˚C (23.35–27.5˚C) for *S. haematobium* (**Fig 3**C and 3D). The thermal breadths ($CT_{min}$–$CT_{max}$) are (16.5–29˚C) for *S. mansoni* and (15–31.2˚C) for *S. haematobium*.

We compared the ability of our $R_0$ thermal curve to predict GNTD non-zero prevalence data compared to a previously published $R_0$ estimate [27] (previous models did not estimate

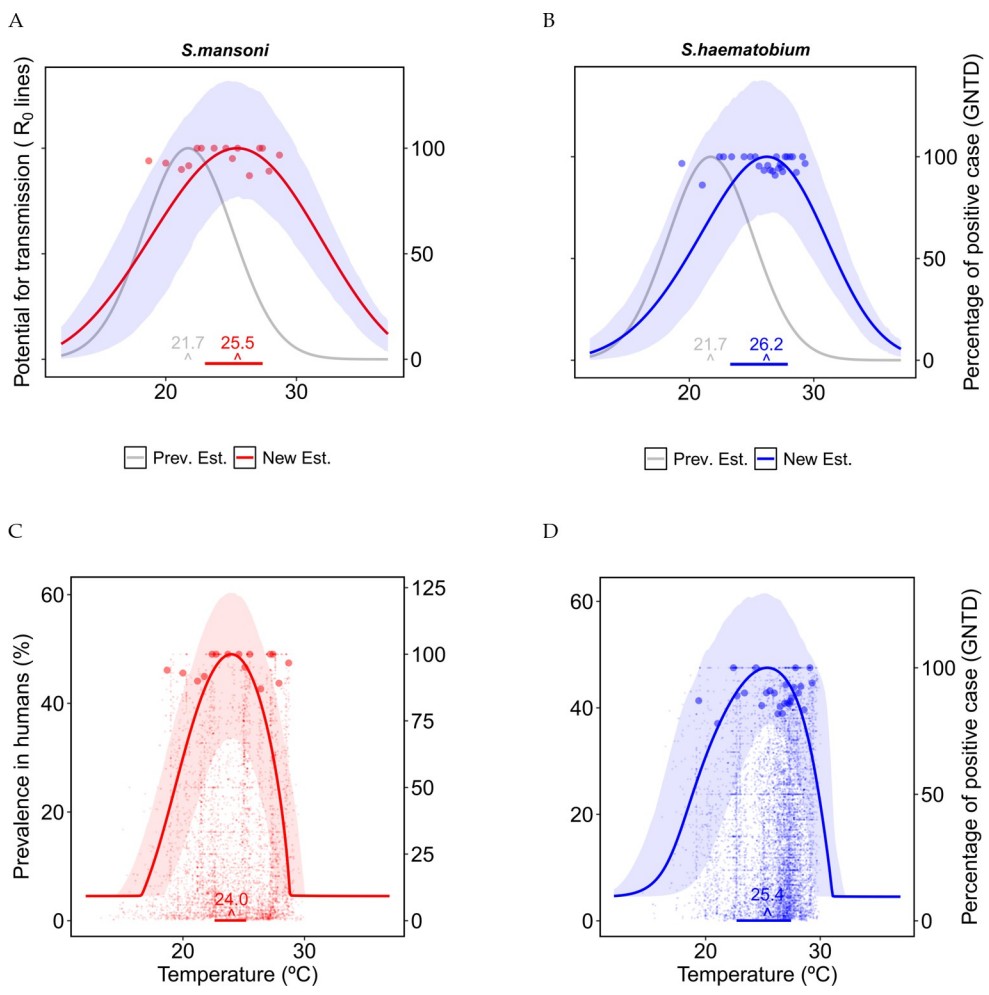

**Fig 3.** Thermal response curve of disease transmission ($R_0$) (A and B) and prevalence of schistosomiasis in humans (C and D) for *S. mansoni* (A and C) and *S. haematobium* (B and D). The shaded areas around the curves are the confidence interval for the TPCs obtained from bootstrap resampling. The small points in C and D represent all data points from GNTD, while each one of the larger dots in A, B, C, and D represents the 98[th] percentile of GNTD prevalence data within each bin of 400 sequential temperature points from the GNTD locations. The gray curve shown in A and B is the previous estimation by [27]. Horizontal lines at the bottom show the 95 percent credible interval of the thermal optimum.

prevalence directly). Using a quantile regression, we found our estimate of $R_0$ is a significant predictor of prevalence for all quantiles ($\tau$ = 0.25,0.5,0.75, 0.9, 0.95, 0.98, p-value < 0.005) while the previous was not a significant predictor of prevalence, except for the 25th quantile (see S1 Supplementary Materials).

## $R_0$ and the derivative across Africa

When mapping temperature-dependent $R_0$ across Africa, we found that most of the continent has mean annual temperature well within the thermal breadth of both *S. mansoni* and S. haemtobium, with the exception of the the upper Sahel desertic and south regions (Fig 4A and 4B). Countries in the northwest of Central Africa have favorable temperatures for both *S. mansoni* and *S. haematobium*. As the tempearture is already close or above the thermal optimum, with increasing temperatures due to climate change, *S. haematobium* prevalence is expected to

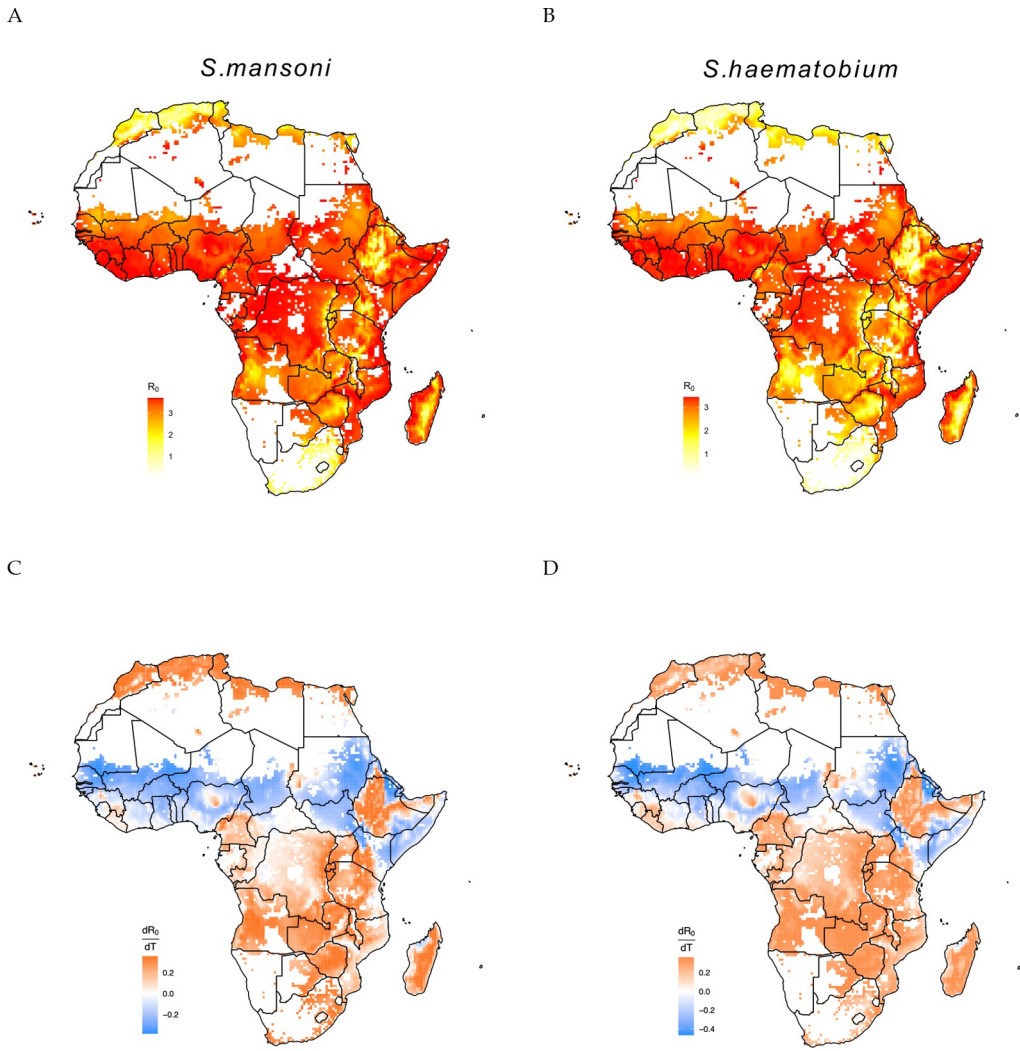

**Fig 4.** A and B show the temperature suitable regions for the transmission of *S. mansoni* and *S.haematobium*, red color refers to the intensity of transmission. The legends show the intensity of $R_0$ based on temperature. C and D show the derivative of $R_0$ respect to temperature for *S.mansoni* and *S.haematobium*, blue and orange colors refer to the regions decreasing and increasing the risk of transmission. The base layer of the map can be found https://www.naturalearthdata.com/downloads/.

decrease more significantly in these regions compared to *S. mansoni*. Fig 4C and 4D show that most of the African regions in the southern hemisphere are characterized by temperatures below the thermal optimum and, therefore, the projected increase in temperature as a consequence of climate change might increase schistosomiasis transmission risk. Conversely, countries in the sub-Sahel regions have temperatures near or above the thermal optimum, so transmission risk may decrease if temperature increases. Further analyses also show that 62.7% of the areas where temperatures are currently suitable for urogenital schistosomiasis transmission are below the thermal optimum and, thus, a small increase in temperature, as predicted by climate change, might lead to an increase in transmission risk (56.2% for intestinal schistosomiasis). Approximately 29.3% of these areas are above the thermal optimum and thus, a temperature increase would lead to a reduction in transmission risk for urogenital schistosomiasis (34.7% for intestinal schistosomiasis), while around 8% of the areas (9.1% for intestinal

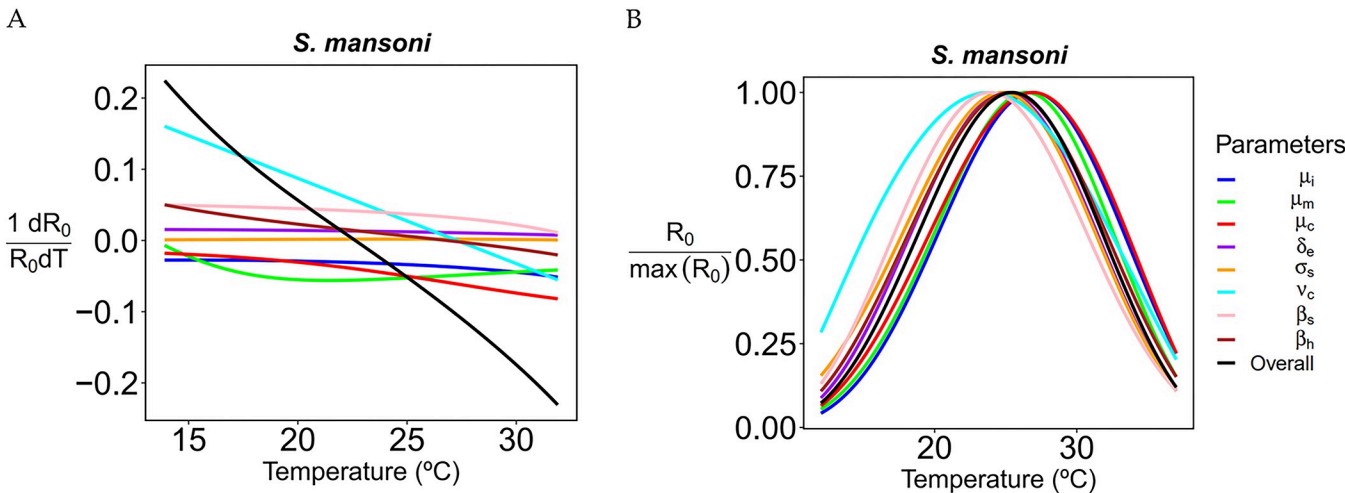

**Fig 5.** The plot on the left side (A) shows the derivative of $R_0$ with respect to temperature per unit of $R_0$ while assuming all parameters (except the focal one) are constant. The derivative is shown for different values of the reference temperature. The plot on the right side (B) shows the behavior of $R_0$ assuming the specified parameter is constant.

schistosomiasis) are close to the thermal optimum and, thus, small changes in temperature are expected to have a negligible effect of on transmission risk. Areas close to the thermal optimum might still experience a decrease in transmission risk if temperatures increase substantially. Ultimately, further simulations of transmission risk will be necessary to precisely assess where and how much transmission will change under alternative climate change scenarios.

## Parameter sensitivity of $R_0$

We performed a sensitivity analysis of $R_0$ with respect to each parameter (Fig 5A and 5B). We found that several demographic and epidemiological parameters have similar impact on $R_0$ around the optimal temperature. However, number of cercaria release rate $v_c(T)$ and mortality rate of miracidia $\mu_m(T)$ have the highest positive and negative impact at low temperatures, respectively, whereas transmission rate in snails $\beta_s(T)$ and mortalitiyt rate of cercaria rate $\mu_c(T)$ have the highest positive and negative impact at high temperatures, respectively.

If we assume that the infected snail's mortality, or mortality rate of the cercariae are constant then, thermal optimum shifts to the right by 1.4°C at 26.9°C. If we assume that the cercarial shedding rate is constant, the optimal temperature shifts toward lower temperatures by 1.7°C is at the lowest value (23.8°C) and shift occurs toward the left. All the results of shifts and their directions are given in Table 2.

**Table 2.** The table shows the magnitude and direction of shift for optimal temperature when we assume that the corresponding parameter is constant.

| Parameter | The shift in opt. temp. (New Opt. temp.) | Shift direction |
|---|---|---|
| Infected snail mortality rate, $\mu_i(T)$ | 1.4°C (26.9°C) | Right |
| Miracidia mortality rate, $\mu_m(T)$ | 0.9°C (26.4°C) | Right |
| Mortality rate of cercariae, $\mu_c(T)$ | 1.4°C (26.9°C) | Right |
| Probability of hatching success of miracidia, $\delta_e(T)$ | 0.5°C (25°C) | Left |
| Transition rate of snails to prepatency, $\sigma_S(T)$ | 0.7°C (24.8°C) | Left |
| Number of cercariae released, $v_c(T)$ | 1.7°C (23.8°C) | Left |
| Transmission rate to snails, $\beta_S(T)$ | 1.6°C (23.9°C) | Left |
| Transmission rateto humans, $\beta_h(T)$ | 0.3°C (25.2°C) | Left |

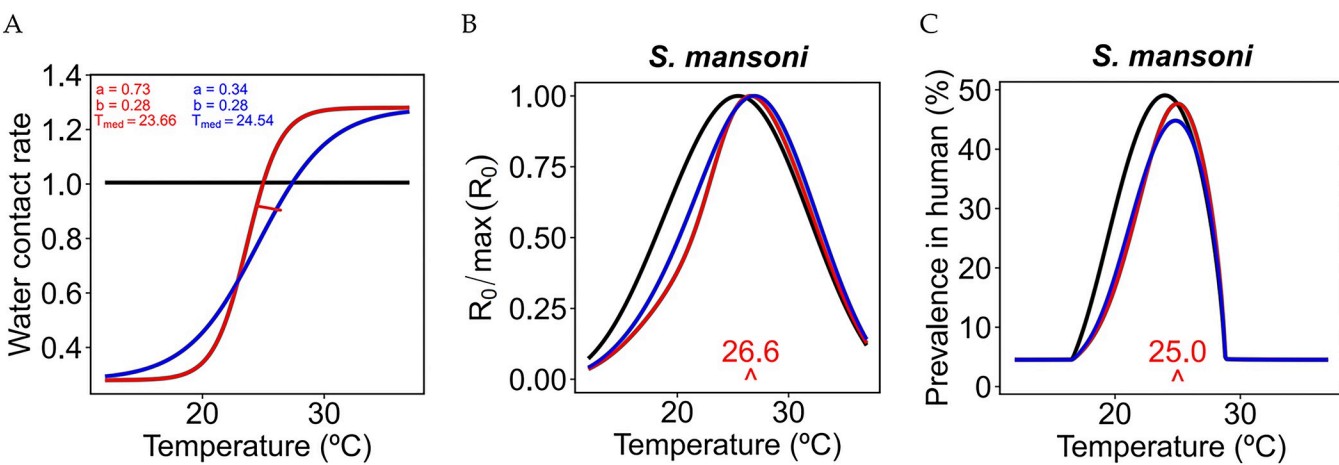

**Fig 6.** Effect of a temperature-dependent water contact rate on the model results. (A) Water contact rate as a function of temperature for different values of the parameters in equation (Eq 9). The red line is the sigmoid function representing the frequency of water contact from two years of records in a village near the Senegal River given by [45]. The effects on $R_0$ and the prevalence in humans are shown in (B) and (C).

### Thermal response of transmission with human water contact rate

We found that the thermal optimum of prevalence in humans and $R_0$ further shifts toward higher temperatures when we accounted for the temperature-dependence of the water contact rate (**Fig 6B and 6C**). We found that both the steepness and $T_{med}$ determine the direction and intensity of the shift. Specifically, we observed a shift to the right larger than 1˚C in the thermal response curve of $R_0$ (**Fig 6B**) and a shift to the right of about 1˚C for the prevalence of schistosomiasis in humans (**Fig 6C**). This analysis was performed only for the *S. mansoni-Biomphalaria* spp. system, as more data were available for their TPC estimation.

## Discussion

We parameterized a detailed mechanistic, process-based, thermal sensitive model accounting for the complex life cycle of *Schistosoma* parasites with the results of an extensive literature review on the thermal tolerance and optima for the LHTs of the parasites *S. mansoni* and *S. haematobium* and their associated snail hosts, which in this context had not been studied separately yet.

According to our analysis, the reproduction number $R_0$ peaks at 25.5 and 26.2˚C for *S. mansoni* and *S. haematobium*, respectively (**Fig 3A and 3B**). Thermal optima derived in our study are thus 4 degrees higher than previously estimated by [27], i.e., 21.7˚C (95% CI 20.5, 23.1) as well as another study [40] which found the thermal optimum of schistosomiasis transmission in calm waters at 16–18˚C, and in flowing waters at 20–25˚C. By accounting for the uncertainty in parameter estimation, we found that the 95% credible intervals of optimal temperature are [23.1, 27.3] and [23.6, 27.9]˚C for *S. mansoni* and *S. haematobium*, respectively. Therefore, even the lower bounds of the 95% credible intervals are equal to or even above the upper estimates by [27]. Our study also found that if the people's contact rate with potentially contaminated water increases with temperature, the thermal optima for both $R_0$ and the prevalence of schistosomiasis in humans would further shift towards higher temperatures. We also found that the thermal optima for $R_0$ occur at slightly different temperatures than the thermal optima for the infection prevalence in humans. Our model projections of the thermal optima for *S. mansoni* and *S. haematobium* fit the GNTD prevalence data better than previous modelling analyses. In addition, the GNTD data suggest that *S. haematobium* may have a larger

thermal breadth than *S. mansoni* (**Fig 3**C and 3D); our model projections better fit this pattern as well.

The model presented here differs in many details from previous thermally sensitive models of schistosomiasis: for instance, we explicitly accounted for a temperature-dependent prepatent period in snails, i.e., a time lag between the infection of a snail and the subsequent beginning of cercarial shedding. In fact, the duration of prepatency is well known to be a decreasing function of temperature, with prepatent periods decreasing by 80% and 75% for *Biomphalaria* and *Bulinus* species, respectively, when temperature increases from 15 to 35˚C (**2**G and 2H). This parameter clearly affects the thermal performance of schistosomiasis transmission, as increasing temperatures dramatically speed up the pace at which snails start shedding cercariae, the infectious stage for humans. Faster transmission rates at higher temperature contribute to shifting the thermal optimum for $R_0$ to higher temperatures with respect to what previously estimated. In addition, we found multiple empirical studies [28,29,74] on the thermal response of egg production by snails, egg hatching rate and hatching time, survival and maturation rates of immature snails, which had not previously been incorporated into models. Increasing temperatures increase the fraction of snail eggs that hatch and reduce the hatching time, thus increasing the pace of transmission, and contributing to further shifting the thermal optimum of prevalence in humans and $R_0$ toward higher temperatures. At some point, however, the benefits for parasite transmission derived from increasing snail and parasite hatching rates at increasing temperatures are more than compensated by the costs due to the exponential increase in snail mortality at temperatures approaching or exceeding 30˚C, thus leading to a sharp drop in $R_0$ in correspondence of the upper thermal threshold.

A higher thermal optimum has implications for the response of schistosomiasis to climate change [3,75–77]. The average temperature in many countries currently affected by schistosomiasis is 22–30˚C [76], spanning below and above the estimated thermal optimum, and thus the response of schistosomiasis transmission to global warming depends on whether temperature increases will lead to approaching or exceeding the thermal optimum. Our study shows that schistosomiasis transmission risk might increase in most of the southern hemisphere regions as a consequence of global warming.

The modelling analysis presented in this paper has several caveats and limitations. Although we incorporated as many of the relevant aspects of schistosomiasis transmission as possible, the complexity of the parasite's life cycle, the variability of LHTs across different species, and the scanty data availability limit our ability to fully explain the thermal dependence of schistosomiasis transmission. The scarcity of species-specific experimental data to describe how temperature affects LHTs, epidemiological parameters and their parasites prevented us to estimate species-specific thermal response curve of $R_0$ and the LHTs curves. To offset the paucity of data over a large thermal breadth, we had to pool together data from snails belonging to the same genus but to different species. Combining the species belonging to the genera *Biomphalaria* and *Bulinus*, respectively, introduces possible sources of uncertainty and bias in the estimation accuracy due to the sparsity of data. Data for some LHTs and model parameters are highly variable, and it is unclear whether this reflects within-species differences in adaptation, or between-species differences in LHTs and epidemiological response, as well as between-study differences in the acclimation protocols and snail husbandry. In addition, for any specific model parameter, we have bootstrapped all data available without stratifying them by published paper, experiment, or snail species; future analyses should thus consider the application of more sophisticated Bayesian approaches. Although we used all the data, we have been able to retrieve for the parameterization of temperature dependence in both schistosomes and snails, our study still has some data limitations. When data were simply not available, as in the case of the cercariae release rate, the cercariae mortality rate, the hatching rate of miracidia,

and the transmission rate in humans for the *S. haematobium-Bulinus* spp. system, we used TPCs estimated for the S. *mansoni-Biomphalaria* spp. system [6]. Therefore, our projection for the *S. haematobium-Bulinus* spp. system is inevitably biased by this choice. Further empirical studies are needed to derive TPCs for all the relevant parameters and the specific snails of public health importance, especially the mortality rate of cercariae and miracidia and the transmission rate in humans for *S. haematobium*. For the same reason, it was not possible to account for potential local adaptation of populations of snails to different climates. Another caveat of our study is that the lack of simpler functions within the rTPC package. Most functions used within the rTPC package are quite complex, which does not match well to the inherent of LHTs. Using simpler (e.g., monotonic) functions might instead help to narrow the confidence intervals associated with the estimation of thermally sensitive parameters. Thus, packages like rTPC need to be improved with simpler and monotonic functions.

The prevalence of schistosomiasis from the GNTD database revealed a high variability across the temperature range that is not fully captured by our mechanistic model. There are many ecological, environmental, and human and snail behavioral factors that influence transmission, which we do not consider here, and which could contribute to that variability. For instance, repeated rounds of mass drug administration are expected to affect prevalence of infection in the human population, as reflected by the GNTD prevalence data, but our analysis did not explicitly account for health policy intervention, an additional factor potentially explaining the variability of GNTD prevalence data that our model is unable to capture. We also derive the thermal response of the basic reproduction number $R_0$ under the assumption of constant temperatures and the modeling approach presented here did not account for the effect of seasonal fluctuations, nor for daily or inter-annual variability of temperature. As an example, seasonal heat waves may induce behavioral responses in snails like diving deeper in search of temperature refugia, or entering an hypobiosis state (aestivation) by digging into the mud, which is a strategy observed for instance in the case of *Bulinus* spp. snails. Future analysis should thus account for the role possibly played by thermal refugia, overwintering, and aestivation shaping the transmission patterns of schistosomiasis. In addition, temperature might affect the dynamics of submerged vegetation where competent snails thrive, which in turn might affect snail abundance and reproductive rates in ways that are not accounted for in our model. Moreover, the thermal response in this study has been described with reference to average water temperatures which also assume to be equal to mean annual temperature. Yet, water temperature is influenced by multiple factors, including depth, geographic location, seasonality, and time of day. Consequently, our estimates may deviate from observed prevalence of infection in a specific location.

Further details, such as snail size [29], which can dramatically affect the thermal response of schistosomiasis, should also be taken into account. Finally, we focus here on temperature as the primary influencing factor in our model. The proposed disease transmission-related dynamics are likely realistic over a broad geographic range, but several other socioeconomic (poverty, access to clean water, etc.), behavioral, climatic (such as precipitation) and environmental (water availability and water quality) factors drive these processes as well [78] and will need to be considered in future analyses to better understand and project how large-scale temperature change will impact schistosomiasis transmission.

## Conclusions

Environmentally mediated diseases often have complex transmission cycles and may respond in nonlinear ways to temperature change. Here, we built a temperature-sensitive model of schistosomiasis based on two common *Schistosoma* parasites in SSA and found a higher

thermal optimum than previously estimated. These results are consistent with prevalence data across Africa and highlight the importance of high-quality and species-specific data on the temperature-dependence of LHTs to project the impact of climate change on disease transmission. Compared to previous models, our analysis reveals a wider range of areas at increasing risk of schistosomiasis transmission mostly in southeast Africa as a consequence of future climate change.

## Supporting information

**S1 Supplementary materials. Fig A: Fecundity rate of snails and confidence intervals** *Biomphalaria* **Spp. Fig B: Fecundity rate of snails and confidence interval** *Bulinus* **Spp. Fig C: Mortality rate of snails and confidence interval** *Biomphalaria* **Spp. Fig D: Mortality rate of snails and confidence interval** *Bulinus* **Spp. Fig E: Prepatent period of snails and confidence interval** *Biomphalaria* **Spp. Fig F: Prepatent period of snails and confidence interval** *Bulinus* **Spp. Fig G: Probability of hatching success of miracidia and confidence interval** *S. mansoni*. **Fig H: Cercariae release rate and confidence interval** *Biomphalaria* **Spp. Fig I: Mortality rate of miracidia and confidence interval** *S. mansoni*. **Fig J: Mortality rate of miracidia and confidence interval** *S. haematobium*. **Fig K: Mortality rate of cercaria and confidence interval** *S. mansoni*. **Fig L: Mortality infected snails and confidence interval** *Biomphalaria* **Spp. Fig M: Mortality infected snails and confidence interval** *Bulinus* **Spp. Fig N: Transmission rate in snails and confidence interval** *Biomphalaria* **Spp. Fig O: Transmission rate in snails and confidence interval** *Bulinus* **Spp. Fig P: Transmission rate in humans and confidence interval** *Biomphalaria* **Spp. Fig Q: Simulation of basic reproduction number for** *S. mansoni* **(left) and** *S. haematobium* **(right). Fig R: Quantile regression of prevalence over** $R_0$ **estimates for our estimate and for Nguyen et al. estimate [27]. Fig S: Blue and red color show when the water contact rate is non constant function of temperature and black color shows when the water contact rate is constant and equals to 1 for all temperature. Table A: The parameters' description. Table B: The selected curves for** *S. mansoni* **and** *Biomphalaria* **species. Table C: The selected curves for** *S. haematobium* **and** *Bulinus* **species.**
(DOCX)

## Acknowledgments

The authors are immensely grateful to Dr. Penelope Vounatsou (Swiss TPH), the curator of the GNTD dataset for their insightful feedback that helped to significantly improve the paper. GADL and IHA are very grateful also to Mark W Denny and Abby McConnell for their valuable suggestions on the early version of the manuscript. This paper is dedicated to the memory of Andrew Brierley.

### Disclaimer/publisher's note

## Author Contributions

**Conceptualization:** Ibrahim Halil Aslan, Lorenzo Mari, Kamazima M. Lwiza, Erin A. Mordecai, Ao Yu, Raquel Gardini Sanches Palasio, Antônio M. V. Monteiro, Eliezer K. N'Goran,

Nana R. Diakite, Mamadou Ouattara, Marino Gatto, Renato Casagrandi, David C. Little, Reed W. Ozretich, Rachel Norman, Fiona Allan, Andrew S. Brierley, Ping Liu, Thiago A. Pereira, Giulio A. De Leo.

**Data curation:** Andrew J. Chamberlin.

**Formal analysis:** Ibrahim Halil Aslan, Julie D. Pourtois, Kaitlyn R. Mitchell, Lorenzo Mari, Kamazima M. Lwiza, Chelsea L. Wood, Erin A. Mordecai, Roseli Tuan, Giulio A. De Leo.

**Funding acquisition:** Giulio A. De Leo.

**Investigation:** Roseli Tuan, Giulio A. De Leo.

**Methodology:** Ibrahim Halil Aslan, Erin A. Mordecai, Giulio A. De Leo.

**Project administration:** Giulio A. De Leo.

**Resources:** Ibrahim Halil Aslan, Giulio A. De Leo.

**Software:** Ibrahim Halil Aslan.

**Supervision:** Lorenzo Mari, Kamazima M. Lwiza, Erin A. Mordecai, Antônio M. V. Monteiro, Eliezer K. N'Goran, Giulio A. De Leo.

**Validation:** Ibrahim Halil Aslan, Julie D. Pourtois.

**Visualization:** Ibrahim Halil Aslan, Julie D. Pourtois, Kamazima M. Lwiza, Giulio A. De Leo.

**Writing – original draft:** Ibrahim Halil Aslan.

**Writing – review & editing:** Ibrahim Halil Aslan, Julie D. Pourtois, Kaitlyn R. Mitchell, Lorenzo Mari, Kamazima M. Lwiza, Chelsea L. Wood, Erin A. Mordecai, Roseli Tuan, Antônio M. V. Monteiro, Devin Kirk, Tejas S. Athni, Susanne H. Sokolow, Eliezer K. N'Goran, Nana R. Diakite, Mamadou Ouattara, Marino Gatto, Renato Casagrandi, David C. Little, Reed W. Ozretich, Rachel Norman, Fiona Allan, Ping Liu, Thiago A. Pereira, Giulio A. De Leo.

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
