## [Decision Letter · Decision Letter 0]

30 Jan 2024

Dear Dr. Aslan,

Thank you very much for submitting your manuscript "Re-assessing thermal response of schistosomiasis transmission risk: evidence for a higher thermal optimum than previously predicted" for consideration at PLOS Neglected Tropical Diseases. As with all papers reviewed by the journal, your manuscript was reviewed by members of the editorial board and by several independent reviewers. In light of the reviews (below this email), we would like to invite the resubmission of a significantly-revised version that takes into account the reviewers' comments. 

We cannot make any decision about publication until we have seen the revised manuscript and your response to the reviewers' comments. Your revised manuscript is also likely to be sent to reviewers for further evaluation.

Sincerely,

Luc E. Coffeng, MD PhD

Academic Editor

Victoria Brookes

Section Editor

Reviewer's Responses to Questions

**Key Review Criteria Required for Acceptance?**

**Methods**

-Are the objectives of the study clearly articulated with a clear testable hypothesis stated?

-Is the study design appropriate to address the stated objectives?

-Is the population clearly described and appropriate for the hypothesis being tested?

-Is the sample size sufficient to ensure adequate power to address the hypothesis being tested?

-Were correct statistical analysis used to support conclusions?

-Are there concerns about ethical or regulatory requirements being met?

Reviewer #1: Appropriate throughout - no concerns

Reviewer #2: The objectives of the study are clearly explained in the introduction and the study design is appropriate to address these questions. However, some more details on the methods used for the literature review are welcome that gives the reader an idea of the effort that has been put in this review e.g. the keywords used, search engines, how many studies were retrieved and how many were deemed useful etc.

Reviewer #3: The objective of the study is clear and a relevant contribution to the field. Generally speaking, the methods are appropriate, but I have several issues that should be addressed before publication.

First of all, it is standard practice to make code and data available and reproducible. The code used in the analysis is not available, as far as I can see. The data on life history traits is only available in a large number of tables in the supplementary materials, which is not an easily accessible format. It is entire unclear what data was used to calculate R0 in the locations in the GNTD database (lines 214-221). To rigorously review the methods, the code necessary to reproduce the analysis should be made available. (I understand this may not be entire possible for the parts of the analysis that involve the GNTD database).

The fitting of thermal performance of the traits is key to the entire analysis. The choice for which functions you chose as the best fit could be better argued for. According to the supplementals, the choice for curves was based on 1) AIC values, 2) the number of parameters, and 3) biological applicability. However, you do not always choose the curve with the lowest AIC value, the number of parameters is already accounted for in the AIC, and you could argue better for what biological applicability is. E.g. you might a priori reject certain curves for certain traits.

The rTPC package is great, but it has limitations, which you do not discuss. Some widely used curves are not part of the package, such as the Growing Degree Day or linear function. I appreciate that you estimated the uncertainty of the estimates - however, resampling assumes that the observations are independent, which the observations here are not (especially where multiple species are pooled together). An alternative approach would be to use Bayesian inference, which multiple of the coauthors on the paper have used in similar studies in the past.

I appreciate how you go about modelling the water contact rate, i.e. first as a constant and then as temperature-dependent.

Showing the disease-free equilibrium (eq. 11, line 201) may be superfluous. I don't see it used anywhere in the results section. 

When comparing to the GNTD data, it could be interesting to know if modeled prevalence was a good predictor of observed prevalence. Right now you only show that R0 is a good predictor of prevalence, but one might expect modeled prevalence would perform even better.

Furthermore, you mention that you limit the analysis to locations with non-zero prevalence (line 216). Temperatures might be one of the reasons why schistosomiasis is absent from these locations, so excluding them could clearly affect your results. Please explain more and at least mention how many data points were excluded in this step.

**Results**

-Does the analysis presented match the analysis plan?

-Are the results clearly and completely presented?

-Are the figures (Tables, Images) of sufficient quality for clarity?

Reviewer #1: Generally clearly presented - some suggestions made on attached comments

Reviewer #2: Some of the TPCs show some counterintuitive shapes (e.g. the death rate curves in fig 2 C and D) How is it possible that the death rate barely increases with decreasing temperature? Is the tolerance limit of the snails this low that they can survive temperatures lower than 10 degrees and isn’t there any data available for lower temperatures? Anyway it is good to use the inset figures for the parameterisation of the model with temperatures under 33 °C

Reviewer #3: The results are convincing and for the most part clearly presented.

I am in particular concerned about the fit for the infected snails' mortality rate, panels 2E-F, which seems to fit the data poorly at the high end. I am wondering if fitting to the lifespan (one over the death rate) would give a better fit? 

Furthermore, fitting the prepatent period as part of a gaussian function (line 269, panels 2G-H) seems like a curious choice. See my comment above about the growing degree day model.

I find the description of figure 3 unclear. I cannot understand what the larger dots are I presume figure 3C-D show modelled prevalence, but this is not clear from the figure description. In figure 3A-B, the interval goes above 1.0, so I assume you divided the R0 from each sample with the mean max(R0). Wouldn't it be more appropriate to divide the values of each sample with the max(R0) from that particular sample, such that R0/max(R0) is strictly below 1?

On line 295, I don't really see how a Gaussian curve should have a Tmin and a Tmax, and if it does, these should be symmetrical around the peak.

**Conclusions**

-Are the conclusions supported by the data presented?

-Are the limitations of analysis clearly described?

-Do the authors discuss how these data can be helpful to advance our understanding of the topic under study?

-Is public health relevance addressed?

Reviewer #1: The conclusions seem valid, and are appropriately nuanced by the authors.

Reviewer #2: Some of the conclusions are a bit too strongly formulated and not fully supported by the data. Especially the conclusion in the abstract on L37: Is there a greater risk of increased transmission with future warming? Were there any projections made in geographic space? What is the statement that there will be a greater risk of increased transmission with future warming based upon? I can imagine this does not hold for all areas, especially those that are already experiencing high temperatures.

Reviewer #3: The results are discussed well, including the wider context, the limitations of the study, and avenues for further research.

The analysis focuses on one prior estimate of optimal temperatures for schistosomiasis transmission (namely reference 25). It would be good to at least mention other estimates as well, such as by McCreesh & Booth (2014).

At the very end of the study (line 420) you mention several countries in Southeast Africa. However, it is unclear to me how you reach this conclusion, as this is the first time in the entire study that your results are discussed in geographic space.

**Editorial and Data Presentation Modifications?**

Reviewer #1: a few questions have been posed and a couple of modification to reduce the amount of cross-checking the reader must do have been made.

Reviewer #2: Introduction:

L57-60: restructure this sentence a bit to the end of the discussion (“we use a mechanistic, process… on the transmission risk of schistosomiasis”) and the next paragraph on schistosomiasis (“a water-associated parasitic disease affecting more than two hundred million people worldwide (not new infections per year!), the vast majority living in SSA”)

L64-66: this is quite a complex sentence, can it be subdivided into smaller sentences?

L67: eggs only hatch upon freshwater contact

L70: this sentence is not completely correct. Now it reads as if people contact the water by burrowing through their skin.

L75: Is there no data on cercariae infectivity or the incubation period of miracidia?

L83-84 which temperature-dependent LHTs have been omitted in papers 37 and 39?

Materials and methods

L149 move the definition of sigma higher so all parameters are explained under the equation that first mentions them. 

L158 Could the authors provide some more details on the methods for the literature review? E.g. the keywords used, search engines, how many studies were retrieved and how many were deemed useful etc. 

L175 MPB has not been defined yet (I assume that MPB is the mean parasite burden in the populations, i.e. the prevalence in the populations?). Additionally, is there any data available on the mean number of parasites in a single human host? I might have missed it but it seems that this parameter is missing although the number of eggs produced per worm inside the human body is present in the equations (ve in equation 1). Or does the model consider only one parasite per human? If so mention this explicitly.

Results

L261 to 267 specify in the text that no optimum for survival could be determined. Otherwise, it is a bit strange to report it for the healthy snails but not for infected snails as this would be a really interesting comparison.

L 305 what is meant by 3.68 and 3.44 peak value? The curve only goes until 1.

Discussion

L348 the authors didn’t look exclusively at thermal tolerance but also at the thermal optima.

Conclusion

L420 Zambia instead of Zombie?

Reviewer #3: Table 1 would be clearer if the first column were split in two: one with the symbols used for the parameters and one with the description of what the parameter is, just as in Table 1 of the supplemntals. 

There are several minor typos in throughout the text. I will mention some:

- The abbreviation MPB is used several times but never written in full

- Line 306, I think 24 C should be 24.0 C. If this is a mistake it occurs several times in the text

- Line 329, I think the word 'the' is missing

- Line 330 and 331, 'reachs' should be 'reaches'

- Line 362, 'reproduction' should be 'production'. 

- Line 377 'Snails' should be 'snails'

- Line 100, my guess is 'to a higher degree' should mean 'to a higher temperature' 

- In the supplementary materials, on page 65, some equations are missing

**Summary and General Comments**

Reviewer #1: In general, I am enthusiastic about seeing this paper published as it pulls together a huge amount of information to come up with more realistic estimates for temperature optima for schistosomiasis transmission for the 2 most important schistosomne species.

See attached comments

Reviewer #2: This paper investigates the thermal optimum for schistosomiasis transmission using a mechanistic modelling approach parameterised by experimental life history data obtained through a literature review. This is undoubtedly a valuable effort for schistosomiasis research and risk assessments and investigations on the effects of climate change on schistosomiasis transmission. The work done is of high quality despite the somewhat limited and outdated data available to parameterise the models. As the authors highlight themselves, more recent and reliable data on both snails and parasites is needed. Related to this data deficiency is the main limitation of this study: the authors used data on different snail species to generalise findings across snail species. It would have been more interesting to run the model per snail species as the species differ between regions, which might result in different optimal temperatures for schistosomiasis transmission. These presumptive differences could result in variable forecasts in schistosomiasis transmission under climate change in different areas. Therefore, I believe that some of the statements made on future schistosomiasis transmission are a bit too strongly formulated as it is not sure that climate change will increase transmission in all areas. Projection of the model results to geographic space could already give a better idea of how transmission dynamics could change in the future under different climate change scenarios in different areas.

Reviewer #3: This paper is a really nice contribution to the field that advances our understanding of the effect of temperature on schistosomiasis transmission.

The schistosomiasis transmission model is well designed and the analysis is complete.

I am requesting major revision because the code and data are not available (in an accessible format).

A better discussion of the limitations of the methods used to fit thermal performance curves would improve the paper, and possibly some further analysis on some of the traits would as well. I find figure 3 unclear, and at the very least the caption should explain the figure better.

Some further copyediting is also needed.

PLOS authors have the

---

## [Decision Letter · Decision Letter 1]

14 May 2024

Dear Dr. Aslan,

Thank you very much for submitting your manuscript "Re-assessing thermal response of schistosomiasis transmission risk: evidence for a higher thermal optimum than previously predicted" for consideration at PLOS Neglected Tropical Diseases. As with all papers reviewed by the journal, your manuscript was reviewed by members of the editorial board and by several independent reviewers. The reviewers appreciated the attention to an important topic. Based on the reviews, we are likely to accept this manuscript for publication, providing that you modify the manuscript according to the review recommendations. 

Sincerely,

Luc E. Coffeng, MD PhD

Academic Editor

Victoria Brookes

Section Editor

Thank you for this revision. One of the reviewers identified a few more issues after looking at the code and the new material in the manuscript. Please address/clarify these and resubmit.

**Methods**

-Are the objectives of the study clearly articulated with a clear testable hypothesis stated?

-Is the study design appropriate to address the stated objectives?

-Is the population clearly described and appropriate for the hypothesis being tested?

-Is the sample size sufficient to ensure adequate power to address the hypothesis being tested?

-Were correct statistical analysis used to support conclusions?

-Are there concerns about ethical or regulatory requirements being met?

Reviewer #3: Thanks for addressing the points raised in the first round - in particular for making your code available.

I just stumbled over an odd thing in your code: in some scripts, bins are 300 and in others 400, while in some scripts, you take the 95th percentile and in others the 98th. (this github search shows it nicely: https://github.com/search?q=repo%3Aibrahimhalilaslan%2FThermal_sensitive_schistosomiasis_model+%22bins_maker%28%22&type=code)

Otherwise I have no further comments on the methods, I think they are entirely satisfactory.

**Results**

Reviewer #3: The authors have made some important improvements to the results and discussion compared to the first version. I also thank the authors for responding to my questions even if the text was not changed. I have a few points, mainly about some of the text that has been added since the first version.

The sentence you added on line 290: "Note that (T_min) and (T_max) are assumed to be the temperature values at which the rate is close to zero". What is close to zero? I tried to look for this in the code but couldn't find it. For *Bulinus* fecundity (panel B), you report the Tmin as 10 degrees in the text, but your plot really doesn't show anything that indicates a critical minimum there. In my view, if a curve doesn't have a clear thermal minimum, there's no need to report one. (As an aside, critical thermal minima/maxima are typically abbreviated CTmin/CTmax).

I now understand figure 3, thanks for the clarifications. I understand that the flat top you get by dividing each replicate could be misleading. However, I think presenting it in the way you do here also has some downsides, namely 1) allowing R0/max(R0) to be above 1 is also confusing; 2) you lose the biological meaning of R0 by dividing with the max; and 3) showing points with prevalence relative to the maximum observed prevalence is confusing, especially if the axis is labelled R0/max(R0). I would propose you instead use 2 y-axes, with the left y-axis just R0 (not normalized - I understand one of the new/previous estimates curves would be higher than the other, but that is also a result), and the right y-axis prevalence. You could add a horizontal line at R0=1 to indicate the thermal limits of transmission, i.e. the temperature range between which both models predict stable transmission is possible. It's also not entirely clear to me why you show the 98th percentile of prevalence of each bin in the figure, while the result you report in the text uses the median. This looks even more odd because there seems to be some data articact that means prevalence is capped at 50% (I understand there is nothing you can do about this). Finally, if I understand this figure correctly, the dots in figures A and C should match, but they don't seem to (the dots in figures B and D do match).

I think the map of Africa is a nice addition, and it is much clearer where the climate data comes from. The jump from annual mean air temperature to observed thermal responses in the lab is big, but saying anything about water temperature as experienced by snails is very difficult, so I think that that is fine for the purposes of this paper. Plotting both R0 and its derivative w.r.t. temperature is clever. One suggestion I have is to use a color scale with a break at R0 = 1, so that your map also shows in which areas your model predicts stable transmission is (not) possible.

You also added a paragraph in the discussion on lines 465-470, where you introduce numbers that are not in the results or methods. I think you can have two options here: either remove these lines and keep a more thorough spatial analysis in the paper you mention you are working on, or move these to the Results section and make it clearer what analysis you actually performed. In particular, I think you jump too quickly from the derivative (below/above the thermal optimum) to increasing/decreasing with climate change. A derivate of 0 does not mean the expected change due to climate change would be 'minimal', because the degree of climate change matters. With extreme climate change, even areas with a dR0/dT above 0 could eventually see decreased transmission. You could also add confidence intervals to all numbers you mention based on the bootstrap to show the robustness of these results.

**Conclusions**

Reviewer #3: The conclusions are clear and well backed up by the results presented.

**Editorial and Data Presentation Modifications?**

Reviewer #3: All references seem to be off by 1, at least halfway through the text. Maybe you just forgot to refresh the reference manager?

On like 315, you mention the "Irf hyperbolic function". I think this what is meant here is the Lobry-Rosso-Flandros (LRF) model - which however does not seem to be hyperbolic. I can't find any reference to a "Irf hyperbolic function" anywhere else.

Is the GNTD dataset publicly available? The link in your references leads to a page which asks for a login.

Finally, I would take one more look at the very first sentence of this paper. You mention the climate has warmed by 0.6 degrees - I think this comes from the Arnell paper, but that compares 1981-2010 with pre-industrial. Most datasets show 1.1 or 1.2 degrees of warming to date. You also cite the third IPCC report from 2001 (why not the latest?), and "some climate change scenarios project even faster increases in the future" sounds a bit weak. I know these are details, but a powerful first sentence could improve the entire paper.

**Summary and General Comments**

Reviewer #3: I appreciate the detailed responses to the points raised in the first round of review. The clarity of the paper has been much improved and the discussion sections is is much more complete. Most of my concerns have been addressed. Almost all of my remaining suggestions are about the way data is presented and do not require major modifications to the paper.

Figure Files:

Data Requirements:

Reproducibility:

References

---

## [Editor Report · Decision Letter 2]

23 May 2024

Dear Dr. Aslan,

We are pleased to inform you that your manuscript 'Re-assessing thermal response of schistosomiasis transmission risk: evidence for a higher thermal optimum than previously predicted' has been provisionally accepted for publication in PLOS Neglected Tropical Diseases.

Best regards,

Luc E. Coffeng, MD PhD

Academic Editor

Victoria Brookes

Section Editor

---

## [Editor Report · Acceptance letter]

3 Jun 2024

Dear Dr. Aslan,

We are delighted to inform you that your manuscript, "Re-assessing thermal response of schistosomiasis transmission risk: evidence for a higher thermal optimum than previously predicted," has been formally accepted for publication in PLOS Neglected Tropical Diseases.

Best regards,

Shaden Kamhawi

co-Editor-in-Chief

Paul Brindley

co-Editor-in-Chief
